# GNNGuard: A Fingerprinting Framework for Verifying Ownerships of Graph Neural Networks

Submission Id: 1091*

## ABSTRACT

Graph neural networks (GNNs) have emerged as the state of the art for a variety of graph-related tasks and have been widely commercialized in real-world scenarios. Behind its revolutionary representation capability, the huge training costs also expose GNNs to the risks of potential model piracy attacks which threaten the intellectual property (IP) of GNNs. In this work, we design a novel and effective ownership verification framework for GNN called *GNN-Guard* to safeguard the IP of GNNs. The key design of the proposed framework is two-fold: graph fingerprint construction and robust verification module. With GNNGuard, a GNN model owner can verify if a deployed model is stolen from the source GNN simply by querying with graph inputs. Besides, GNNGuard could be applied to various GNN models and graph-related tasks. We extensively evaluate the proposed framework on various GNNs designed for multiple graph-related tasks including graph classification, graph matching, node classification, and link prediction. Our results show that GNNGuard can robustly distinguish post-processed surrogate GNNs from irrelevant GNNs, e.g., GNNGuard achieves 100% true positives and 100% true negatives on the test of 200 suspect GNNs of both graph classification and node classification tasks.

## CCS CONCEPTS

• **Security and privacy** → **Privacy-preserving protocols**; • **Computing methodologies** → **Machine learning**.

## KEYWORDS

Graph Neural Networks, Model Intellectual Property Protection, Model Fingerprinting

**ACM Reference Format:**

Anonymous Author(s). 2018. GNNGuard: A Fingerprinting Framework for Verifying Ownerships of Graph Neural Networks. In *Proceedings of Make sure to enter the correct conference title from your rights confirmation emai (Conference acronym 'XX)*. ACM, New York, NY, USA, 19 pages. https://doi.org/XXXXXXX.XXXXXXX

## 1 INTRODUCTION

In the ever-evolving landscape of artificial intelligence and machine learning, the rapid advancements in deep learning techniques have propelled the field into new frontiers of innovation. Among these advancements, Graph Neural Networks (GNNs) have emerged as a potent tool for modeling and analyzing complex data represented as graphs [4, 10–13, 18, 31, 34, 42]. From social network analysis to recommendation systems and drug discovery, GNNs have found widespread applications in diverse domains, revolutionizing the way we understand and manipulate graph-structured data [20, 42]. As the research community and industry sectors invest heavily in the development of innovative GNN models and training techniques, there arises a compelling need to protect these intellectual assets from unauthorized utilization [29], replication [29], or model piracy attacks [3, 22, 24–27, 29, 47], i.e., protecting the intellectual property (IP) of GNNs[1].

Strides have been made to safeguard the intellectual property of deep learning models, where the keystone is to verify the ownership of the model [33, 45]. Given a suspect model, the verification of ownership is to determine whether it is a pirated model or an irrelevant model. Once piracy occurs, the model owner can take follow-up actions, e.g., collecting other evidence and filing a lawsuit [24, 33], to protect their innovations. Generally, existing approaches rely on marking the model ownership by model outputs [24, 33, 37]. If specific inputs can obtain predefined outputs, the suspect model is verified to be a pirated version. Existing ownership verification methods have been effective in traditional deep learning models [32, 33], facing formidable challenges when applied to GNNs due to their unique structure and inherent complexity. In the following, we elaborate on the inherent limitations of applying existing approaches to safeguard GNNs:

- *Model watermarking* approaches mark the model ownership by embedding hidden functionalities into the protected model during training or fine-tuning [37, 45]. This can be achieved by assigning specific labels to predefined watermark data and mixing them into the training dataset [37], backdoor techniques [45] and other adversarial forgery techniques [19]. The model owner can then utilize the watermark data to identify whether a suspected model is a pirated version by matching its prediction results. Recent studies transfer this to mark GNN ownership by graph backdoor attack [37, 45] which negatively impacts the performance of watermarked GNN [46][2]. This is mainly because triggers in terms of subgraphs added to the graph data will influence the prediction of other nodes due to the effect of node interaction and coalition [43].

- *Model fingerprinting* has emerged as a promising approach for intellectual property (IP) protection. Given a protected model, a set of input samples, i.e., *fingerprints*, are found to have specific

---

[1]**Relevance:** Unauthorized use of graph algorithms and infringement on model IP, especially in the context of web data, raises significant ethical and legal concerns. Verifying ownership of GNNs acts as a deterrent against unauthorized usage and makes web-related graph learning applications more robust and secure.

[2]We validate this and report the empirical results in Section A.4.2

outputs that serve as a digital signature of ownership encapsulating its architecture, training graph, and parameters [5, 33]. Unlike watermarking, model fingerprinting does not require any modifications to the model parameters, thus preserving its normal utility, which has become a hot-line of IP protection in recent years [5, 32, 33]. However, existing fingerprinting approaches only consider traditional deep learning models such as image classification and text generation, leaving graph-related tasks untouched. As a result, they hardly approach the multiple downstream tasks and the intrinsic complexity of GNNs. To the best of our knowledge, no effort has been made to tailor the model fingerprinting for GNNs.

To fill this gap, in this work, we propose a unified model fingerprinting framework to verify the ownership of GNNs, which is agnostic to the multiple downstream applications and is efficient in searching for unique fingerprints. Specifically, we present a unified framework of fingerprinting GNNs which is agnostic to multiple downstream tasks, with two essential modules: **Graph Fingerprint** and a **Unified Verification Mechanism** (dubbed Univerifier). With the graph fingerprint as inputs of the suspect GNN, the Univerifier will justify if it is a pirated one according to the concatenated outputs of the suspect GNN. Then, considering that the unique fingerprint of a GNN encapsulates its architecture, training graph, and parameters, we construct a set of GNNs to imitate the behavior of pirated GNNs and irrelevant GNNs by different model obscuring techniques [37, 45]. After this by jointly optimizing the graph fingerprint and the Univerifier on such models, the Univerifier will learn to match the outputs between the protected GNN and the pirated ones and distinguish them from the irrelevant GNNs' outputs, with graph fingerprint as inputs. Considering the discrete input structure of GNNs, we propose an effective optimization strategy to generate a valid graph fingerprint. Especially, after the random initialization of graph fingerprint nodes and adjacencies, they are updated according to the exclusively predicted outputs of the Univerifier and then are projected into the original space respectively. To summarize, our contributions are as follows,

- We propose a novel framework for protecting the IP of GNNs based on model fingerprinting, which could effectively verify the ownership of a GNN without degrading its normal utility.
- One advantage of the proposed framework is task-agnostic. With the aid of the learnable verifier, the framework could be applied to various GNN tasks such as graph classification, node classification, and link prediction.
- We extensively evaluate the performance of the proposed GNN fingerprinting on two graph-level tasks, one node-level task, and one edge-level task. GNNGuard achieve noticeable verification performance on various tasks. For instance, GNNGuard achieves 100% true positive and 100% true negative on both graph classification and node classification tasks.

## 2 BACKGROUND & RELATED WORK

### 2.1 Graph Neural Networks

Let $G = \{V, A, X\} \in \mathcal{G}$ denote a graph with node set $V$, adjacency matrix $A$, and attribute matrix $X$. There are $|V|$ nodes in the node-set $V$ and each node $v \in V$ has an attribute $x \in X \in \mathbb{R}^{|V| \times d}$ and

$d$ is the dimension of the attribute. Adjacency matrix $A$ contains information of graph topology as $A_{v,u} = 1$ denotes an edge between $v \in V$ and $u \in V$. Graph Neural Networks (GNNs) such as graph convolutional network (GCN) [13] and Graphsage [10] take the graph data as input of which the objective is to capture both the topology and attribute information of graphs. A GNN denoted as $F$ takes an input $I$ and produces an output $O$ such that $F(I) = O$. Specifically, we denote $O_u$ as the output of node-level tasks, $O_g$ as the output of graph-level tasks, and $O_e$ as the output of edge-level tasks. *If not specified otherwise, we utilize O to denote all kinds of GNNs' outputs.* To train GNNs to capture accurate representations at different levels of tasks, both graph data and ground-truth labels of specific tasks will be provided which are denoted as $\mathcal{D}$. Then, using labeled training data $\mathcal{D}$, the parameters of GNN $F$ will be optimized by the loss calculated between predicted results and ground truth. After training, the GNN will be released to users for further utilization [16, 38], or deployed as an online service [16] where users can leverage the learned GNN to obtain predictions in corresponding graph tasks. Existing GNN tasks are categorized into three types:

- **Graph-level tasks.** Graph classification and graph matching are two representative sub-tasks of graph-level tasks. The input of graph classification is a single graph where $I = G$ and the output is the prediction label $O = \hat{y} \in \mathcal{Y}$; whereas the inputs of graph matching are two graphs where $I = (G_1, G_2)$ and the outputs is a value of similarity, i.e., $O = \hat{o} \in \mathbb{R}^1$.
- **Edge-level tasks.** Link prediction is an edge-level task in which the input is a whole graph $I = G$ and the output is a probability matrix $O = \hat{A} \in \mathbb{R}^{|V| \times |V|}$.
- **Node-level tasks.** Node classification takes a whole graph $I = G$ as input and outputs predictive labels for all nodes $O = \hat{Y} \in \mathbb{R}^{|V|}$.

### 2.2 Intellectual Property Protection of Graph Neural Networks

*2.2.1 Intellectual Property Infringement of GNNs.* As Graph Neural Networks (GNNs) are disseminated to users for further utilization or deployed as online services (hereafter referred to as target GNNs), the threat of *model piracy* against GNNs is garnering increasing attention. Malicious attackers may exploit systemic or algorithmic vulnerabilities to steal the parameters or functionality of these target GNNs. For instance, attackers can engage in model piracy through software/hardware vulnerabilities [24, 28], enabling them to pilfer the parameters of the protected model directly from the server owned by the model creators.

Furthermore, given the API service of the target GNNs, malicious attackers can act as normal users and query the API service to implement *model stealing attacks* [28, 30]. The objective of model stealing attacks is to build a surrogate model that matches the accuracy of the target model with fewer training resources. For instance, in studies like [6, 28, 36], attackers prepare original graphs which are sampled from a similar domain with the target GNN's dataset and learn the surrogate GNN with outputs from the target GNN. Unlike conventional model stealing attacks, extracting GNNs presents an additional challenge due to the discreteness involved in constructing query samples. Some studies make assumptions on the attacker's knowledge, e.g., the attacker is able to obtain the

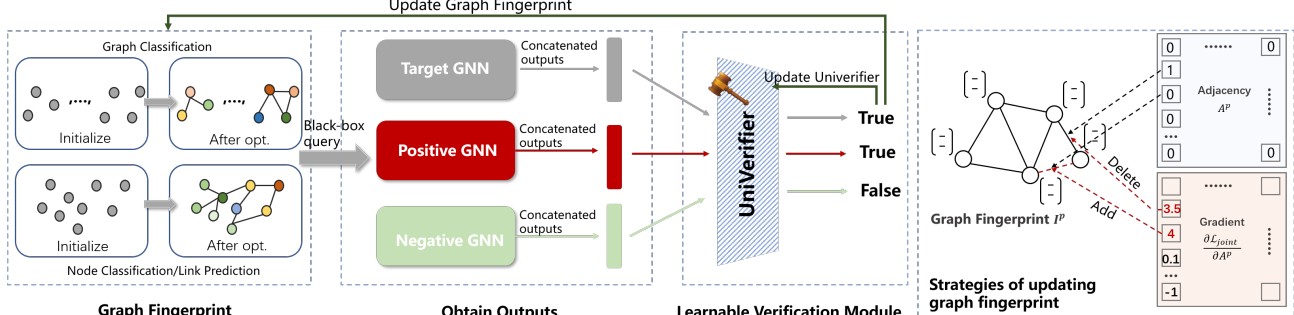

Figure 1: Framework overview of GNNGuard.

whole graph structure or partial structure in node classification tasks [28], and is able to know the attribute set of all nodes [36]. Then use this knowledge to construct graphs for querying. Some works assume that the attackers have no knowledge of the training dataset, and thus utilize graph reconstruction techniques [44] to first construct discrete graph structure and node attributes from the target GNNs and then use them in the following stealing.

Consequently, malicious attackers deploy these pirated GNNs as new services or redistribute them to customers for financial gain [5]. This not only infringes upon the intellectual property rights of GNN owners but also obstructs the cultivation of a responsible AI research and development culture.

*2.2.2 Ownership Obfuscation.* To obfuscate ownership, the attacker would apply a variety of post-processing techniques such as fine-tuning, pruning, and distillation, on the pirated GNNs. We mainly illustrate the following classes of obfuscation techniques that the attacker is likely to apply to obfuscate the ownership of GNNs.

- **Fine-tuning & Partial Retraining:** To obfuscate the ownership of a pirated model, attackers may resume the training data of the stolen model on public data collected from a similar domain of training data, and fine-tune several layers of the model. Specifically, in the settings of GNNs, the attacker can fine-tune $K$ GNN layers according to the learning objective with the other layers fixed. In comparison, during partial retraining, the parameters of selected $K$ layers are first randomly initialized before training is resumed. After this, the pirated GNN can fall into a different local optimum, preserving the original utility, but exhibiting divergent output behaviors [9].
- **Distillation:** Distillation-based obfuscation adopts knowledge distillation techniques by viewing the pirated model as a teacher model and the model of different architecture as a student model. In the settings of GNNs, the learned graph topology information is distilled from the pirated GNN to a new GNN with different architectures, e.g., different graph convolution architectures by matching their outputs. Distillation exacerbates the obfuscation of model ownership due to the transformed model architecture resulting in altered predictive behaviors [33].

*2.2.3 Ownership Verification.* IP protection aims to prevent malicious attackers from infringing the innovations of model owners. IP encompasses various components of a model, including its architecture, training data, and weights, making it inherently complex to represent through conventional signatures. Model ownership verification leverages deep learning models, as their output behaviors

are intricately linked to factors such as architectures, training data, and model weights [5, 15]. As a result, model ownership verification now become the keystone of model IP protection. In the context of suspect models, a model ownership verification mechanism aims to query the model and ascertain whether it is a pirated version or an unrelated GNN, based on its outputs. Existing model ownership verification approaches can be categorized into model watermarking [37, 45] and model fingerprinting [5, 15, 33, 39].

**GNN Watermarking.** Model watermarking involves active modifications to a model's architecture, training data, or weights to elicit specific outputs with predefined input data. However, this inevitably influences the normal utility of the protected model. Very recently, there has been some work discussing model watermarking on GNNs [37], which rely on targeted poisoning attacks against graph classification models to embed watermarks. Such a performance degradation of watermarking is also discovered in watermarked GNNs.

**GNN Fingerprinting.** Therefore, in this paper, we resort to model fingerprinting, which generates specific inputs to query representative outputs and requires no modifications on models. Generally, GNN fingerprinting determines whether a suspect GNN $\tilde{F}$ is pirated from the target GNN in the two stages:

- **Fingerprints Construction.** At this stage, fingerprints encoding the essential characteristics of the target GNN, which is named *graph fingerprint*, are constructed.
- **Fingerprints Verification.** At the verification stage, the suspect GNN is attested via the black-box access, to determine whether and what confidence the graph fingerprint is also present in the suspect GNN. Additionally, considering that GNNs are applied to multiple applications, the verification mechanism should be agnostic to the downstream tasks.

## 3 METHODOLOGY

### 3.1 Overview of GNNGuard

One of the obstacles to fingerprinting GNNs is the various downstream tasks. In this paper, we for the first time propose a unified fingerprinting framework named *GNNGuard* for GNNs. The framework has two essential modules: graph fingerprint and a unified verification mechanism.

- **Graph Fingerprint** is denoted as $\mathcal{I} = \{I^p\}_{p=1}^{P}, I_p \in \mathcal{G}$, which are used to query the suspect GNN and obtain $O = \{O^p\}_{p=1}^{P}$.

- **Unified Verification Mechanism**, is denoted as $\mathcal{V} : O \rightarrow \mathbb{S}^2$ and is implemented by a binary deep neural network (DNN) classifier, named Univerifier. It leverages the concatenated outputs $O$ as input and predicts whether the suspect GNN is pirated or irrelevant. Specifically, the output of the Univerifier is a 2-dimensional probability simplex $\{(o_+, o_-) \mid o_- + o_+ = 1, 0 \leq o_-, o_+ \leq 1\}$. If $o_+ > \lambda$ the suspect GNN is determined as a pirated version where $\lambda$ is a threshold to determine.

As illustrated in Fig 1, the general pipeline of GNNGuard mainly consists of the following three keystones:

- **Key1.** (*Anti-Obfuscation GNN Preparation*) To train Univerifier to distinguish pirated GNNs from irrelevant GNNs, we first craft a set of suspect models based on the target GNN with the aid of public data from the same domain of the owner's training data. After this, the obtained positive and negative models denoted as $\mathcal{F}_+$ and $\mathcal{F}_-$ will be used to search for graph fingerprint and train the Univerifier.
- **Key2.** (*Graph Fingerprint Construction.*) As Fig 1 shows, we define the patterns of graph fingerprint for different levels of GNN tasks, i.e., $\mathcal{I}$. After this, we obtain the initialized graph fingerprint for the following optimization.
- **Key3.** (*Jointly Learning for Graph Fingerprint and Univerifier*).) We jointly optimize the graph fingerprint and Univerifier with the objective: Univerifier can accurately classify the outputs of pirated GNNs and the target GNN as positive ($o_+ > \lambda$) while the irrelevant GNNs as negative ($o_+ < \lambda$).

## 3.2 Anti-Obfuscation GNN Preparation

There are two objectives of ownership verification: (i) *Robustness*: accurately recognize the pirated GNNs. (ii) *Uniqueness*: do not recognize independently trained GNNs as pirated GNNs. However, with only the target GNN, it is difficult to achieve robustness and uniqueness. First, to evade detection, the attackers will post-process the pirated GNNs resulting in significantly different behavioral patterns from the target GNN [24]. Second, the architecture of GNNs is also determined by the training graph, and if the distributions of the training graph used for training irrelevant GNNs are similar to the target GNNs, the behavioral patterns of these GNNs will be similar. To tackle these, we propose to first craft a set of GNNs to imitate the behaviors of different GNNs.

- **Prepare Pirated GNNs, named Positive GNNs**. We derive a representative set of pirated suspect GNNs based on the target GNN by applying the obfuscation techniques mentioned in Section 2.2.2. The obfuscation covers a wide range of hyperparameter configurations. For instance, we apply fine-tuning on different GNN layers and partial retraining with different initialization points. The set of positive GNNs is denoted as $\mathcal{F}_+$ with each suspect GNN $F_+$ in the set being applied one or more obfuscation techniques.
- **Prepare Irrelevant GNNs, named Negative GNNs**. There are two kinds of irrelevant GNNs: (i) GNNs independently trained on different domains; (ii) GNNs independently trained on similar domains. To achieve IP protection, it is reasonable to download a number of pre-trained models from online sources (e.g., Hugging Face and Pytorch Hub) that belong to the first kind of irrelevant GNNs. To produce the second kind of irrelevant GNNs having

similar behaviors to the target GNN, the downloaded models are fine-tuned on the domain-relevant public data (e.g., the subset of training data). We denote the prepared irrelevant suspect GNNs as $\mathcal{F}_-$ and the set contains both kinds of irrelevant GNNs.

After the preparation, the remaining problems are how to leverage the prepared positive GNNs and negative GNNs to optimize the verifier and the fingerprints to achieve robustness and uniqueness.

## 3.3 Graph Fingerprint Construction

In the context of generating fingerprints for GNNs, we divide the generation into two key phases: (i) Initializing; (ii) Optimizing (which is detailed in Section 3.4).

For each graph (or the single graph) of graph fingerprint, there are three components to be initialized: (i) Node set $V^p \in I^p$; (ii) the adjacency matrix $A^p \in I^p$; (iii) Node attributes $X^p \in I^p$. First, we initialize the node set with a uniform number $n$ of nodes. Second, the adjacency matrix is initialized by randomly selecting a very low fraction $r$ of nodes to be linked, i.e., $A^p_{v_i,v_j} \sim \mathbb{B}(1, \epsilon)$. Third, for each node, each dimension of the attribute is uniformly initialized in the ranges. As illustrated in Fig 1(a), we specialize the samples of graph fingerprint to different GNN tasks as follows.

- **Node-level samples.** In node-level tasks like node classification, the output is the predicted vector of an individual node. graph fingerprint is defined as a single graph $\mathcal{I} = \{I\}, I \in \mathcal{G}$. To achieve efficient verification, we sample the outputs of $m$ nodes that are used to verify the ownership, i.e., $O = \{O_n^{v_i}\}_{i=1}^m, v_i \in V$.
- **Edge-level samples.** The outputs of GNNs in these tasks represent edge information. Link prediction is one of the representative edge-level tasks [41]. graph fingerprint is defined as a single graph $\mathcal{I} = \{I\}, I \in \mathcal{G}$, and we sample the output $m$ node pair, i.e., $O = \{O_e^{v_i,v_j}\}, (v_i, v_j) \in E$.
- **Graph-level samples.** In graph-level tasks like graph classification, the output is the predicted vector of the whole graph, while in tasks like graph matching, the output is the matching vector of the whole graph. Graph fingerprint are defined as several graphs $\mathcal{I} = \{I^p\}_{p=1}^P, I^p \in \mathcal{G}$. The outputs used to verify ownership are the prediction results of these input graphs, i.e., $O = \{O_g^p\}_{p=1}^P$.

## 3.4 Jointly Learning for Graph Fingerprint and Univerifier

*3.4.1 Unified Verification Mechanism.* Existing fingerprinting verification mechanisms are often tailored for classification tasks, which focus on matching the output labels of fingerprints with pre-defined labels [33] and are not applicable to various GNN tasks, such as link prediction [41] and graph matching [2]. Different GNN tasks have varying input and output formats, causing significant differences in the definition and value of $O$ on different tasks (Section 2.1). For instance, for node classification task, model outputs are prediction vectors; while for other GNN tasks, such as link prediction [41] and graph matching [2], where the outputs represent continuous similarity values. To this date, we introduce the Univerifier $\mathcal{V} : O \rightarrow \mathbb{S}^2$. Univerifier takes the concatenated outputs of suspect models on fingerprints as inputs, and its output is a probability simplex $\{(o_+, o_-) \mid o_- + o_+ = 1, 0 \leq o_-, o_+ \leq 1\}$. To determine ownership, we only have to set a threshold $\lambda$ for $o_+$. That

is, if $o_+ > \lambda$ the suspect GNN is determined as positive where $\lambda$ is a hyperparameter. With the graph fingerprint and the prepared GNNs, we train the Univerifier with the following objective.

$$\arg\max_{\mathcal{V}} \log o_+(F) + \sum_{F_+ \in \mathcal{F}_+} \log o_+(F_+) + \sum_{F_- \in \mathcal{F}_-} \log o_-(F_-) \quad (1)$$

where $o_+(F_+)$ denotes the prediction result $o_+$ of Univerifier on model $F_+$'s output.

*3.4.2 Joint Learning.* We propose a joint learning framework to learn both parameters of the Univerifier and graph fingerprint. In a word, the objective of the joint learning framework can be formulated as follows.

$$\arg\max_{\mathcal{V}, \mathcal{I}} \mathcal{L}_{\text{joint}} = \sum_{I_p \in \mathcal{I}} \log o_+(F(I_p)) + \sum_{I_p \in \mathcal{I}} \sum_{F_+ \in \mathcal{F}_+} \log o_+(F_+(I_p))$$
$$(2)$$
$$+ \sum_{I_p \in \mathcal{I}} \sum_{F_- \in \mathcal{F}_-} \log o_-(F_-(I_p)).$$

Different from existing studies focusing on the image domain, the adversarial techniques of updating images can not be directly applied to data in graph space [24]. We resort to adversarial techniques to generate adversarial examples of graph data [48]. In the following, we elaborate on how to update the individual graph $I^p$ of graph fingerprint $\mathcal{I}$. Since many graph datasets lack node attributes, our approach focuses on updating the adjacency matrix. If the dataset includes node attributes, we also update the matrix accordingly.

- *Update Adjacency Matrix.* We first propose to compute the gradient of the adjacency matrix according to Eq 2, i.e., $g^p = \nabla_{A^p} \mathcal{L}_{\text{joint}}$. However, the gradient from continuous space can hardly be used to update the discrete adjacency matrix. After this, we propose to discretize the gradient $g^p$. Each entry of gradient $g_{u,v}^p$ represent the significance of edge connecting node $i$ and node $j$ on $\mathcal{L}_{\text{joint}}$. The edge having larger $abs(g_{u,v}^p)$ will have a significant impact on the verification result. Therefore, we rank a set of edges denoted as $E^p = \{e_i^p\}_{i=1}^K$ having the top-$K$ large value of $abs(g_e^p), e = (u, v)$. We use $\tilde{g}^p$ as the new gradient after ranking. To decide whether an edge should be added or deleted, we follow these rules: (i) if the edge $e$ is actually on the graph and $g_e^p \leq 0$ the edge will be deleted; (ii) if the edge $e$ does not exist on the graph and $g_e^p \geq 0$ the edge will be added. The illustration of updating adjacency matrix is presented in Fig 1.
- *Update Node Attributes.* The key to update node attributes is to project the node attributes in their domain. For instance, one of the attributes is the age of user nodes, and after updating the age should be a discrete value. Therefore, before updating, we first collect the ranges or the labels of each attribute which will be used to project the updated node attributes, denoted as $C$. Then we update the attribute matrix $X^p$ as $X^p \leftarrow \text{clip}(X^p + \alpha \nabla_{X^p} \mathcal{L}_{\text{joint}})$. The function clip($\cdot$) is a row-wise operation according to $C$. Specifically, (i) If the attribute is continuous but has a value range we will project the value exceeding the range to the minimum or maximum. (ii) If the attribute is discrete we will project the value to its closest label.

## 3.5 Scalable and Theoretical Guarantee

*3.5.1 Scalable Guarantee.* The training complexity of GNNGuard consists of two main components. First, we consider obtaining the outputs on graph fingerprint. Assuming that we have $N$ graph fingerprint, the complexity of obtaining the outputs for a suspect GNN is $O(CEN)$ where $C$ is a constant and $E$ is the number of edges. Second, we conduct verification using Univerifier. The input dimension of the Univerifier is $Nd$, where $d$ represents the dimension of the GNN output. Univerifier is implemented as a multi-layer fully connected neural network with hidden layer sizes $[d_1, ..., d_\ell]$. Consequently, the time complexity of verification using Univerifier is $O(NdO_M)$, where $O_M = d_1 \times ... \times d_\ell$. Finally, the total complexity of training is $O(CEN_m + NdO_m)$, taking into account the size of the model set $\mathcal{F}_+ \cup \mathcal{F}_-$, the number of edges on graph fingerprint and the number of fingerprints. To scale our framework to large graphs, we recommend reducing the size of the positive/negative GNNs set and the number of graph fingerprint.

*3.5.2 Theoretical Guarantee.* We provide a theoretical analysis of the robustness, i.e., the probability of correctly identifying an indeed stolen suspect GNN as a negative GNN. It is important to note that we focus on the node classification task for the purpose of presenting theoretical analysis[3]. The theoretical guarantees of robustness are presented as follows. First, we introduce the following Lemma from [17].

LEMMA 3.1. *Suppose that a GNN $f_w$, of which the parameters are denoted as $\{W_i\}_{i=1}^\ell$. Pertubations $\{U_i\}_{i=1}^\ell$ are added on $f_w$ to obtain its surrogate GNN $f_{w+u}$ (positive GNN). The differences in outputs of $f_w$ and $f_{w+u}$ given the same input graph can be bounded as follows,*

$$| f_{w+u}(X,A) - f_w(X,A) | \leq eBd_{max}^{\frac{\ell-1}{2}} \left( \prod_{i=1}^\ell \|W_i\|_2 \right) \sum_{k=1}^\ell \frac{\|U_k\|_2}{\|W_k\|_2} = \mathcal{B} \quad (3)$$

*Note that $d_{max}$ is the largest node degree on the input graph.*

We derive the following theoretical guarantees of robustness,

THEOREM 3.2. *Denote that the output distribution of the target GNN is defined as a normal distribution $\mathcal{N}(\mu_0, \sigma^2)$. According to the bound $\mathcal{B}$, the output distribution of a positive GNNs can be defined as $\mathcal{N}(\mu_0 + \mathcal{B}, \sigma^2)$ and $\mathcal{N}(\mu_0 - \mathcal{B}, \sigma^2)$. Then the probability of the differences in outputs of the above two GNNs being smaller than $\lambda$ can be, i.e., the robustness,*

$$P(Diff < \lambda) = \Phi(\frac{\lambda + \mathcal{B}}{\sigma}) + \Phi(\frac{\lambda - \mathcal{B}}{\sigma}). \quad (4)$$

*Note that $\Phi(\frac{x-\mu}{\sigma}) = \frac{1}{\sigma\sqrt{\pi}} exp - \frac{(x-\mu)^2}{2\sigma^2}$.*

The omitted proof is presented in Appendix. The robustness is related to (i) the maximum node degree of graph fingerprint and (ii) the perturbations on parameters. With a larger maximum node degree, the absolute value of $\mathcal{B}$ will be larger which will decrease the probability of recognizing positive GNN resulting in worse robustness. We aim to utilize this to guide the construction of graph fingerprint to have nodes with the largest degrees.

---

[3]Additional analyses on other tasks are provided in the Appendix.

**Table 1: Mean test accuracies of various GNN tasks on different datasets. "-" means not applicable. The best results are in bold.**

| Task | Dataset | Schemes / Model | DeepFool | IPGuard | GNNGuard |
|---|---|---|---|---|---|
| Graph Classification | ENZYMES | GCNMean | 0.711 | 0.507 | **1.00** |
| | | GCNDiff | 0.544 | 0.498 | **1.00** |
| | | GraphsageMean | 0.651 | 0.508 | **1.00** |
| | | GraphsageDiff | 0.757 | 0.493 | **1.00** |
| | PROTEIN | GCNMean | 0.574 | 0.513 | **0.967** |
| | | GCNDiff | 0.597 | 0.503 | **0.961** |
| | | GraphsageMean | 0.581 | 0.5 | **0.989** |
| | | GraphsageDiff | 0.529 | 0.508 | **0.984** |
| Graph Matching | AIDS | GCNMean | 0.894 | - | **0.952** |
| | | GCNDiff | **0.935** | - | 0.925 |
| | | SimGNN | 0.892 | - | **0.943** |
| | LINUX | GCNMean | 0.733 | - | **0.933** |
| | | GCNDiff | 0.74 | - | **0.797** |
| | | SimGNN | 0.653 | - | **0.752** |
| Node Classification | Cora | GCN | 0.895 | 0.478 | **1.00** |
| | | Graphsage | 0.963 | 0.529 | **1.00** |
| | Citeseer | GCN | 0.891 | 0.451 | **0.986** |
| | | Graphsage | 0.957 | 0.483 | **0.992** |
| Link Prediction | Cora | GCN | 0.807 | - | **0.838** |
| | | Graphsage | 0.808 | - | **0.823** |
| | Citeseer | GCN | **0.855** | - | 0.844 |
| | | Graphsage | 0.859 | - | **0.873** |

## 4 EVALUATION RESULTS

In this section, we study GNNGuard by answering the following research questions. **(RQ1)**: How is the verification accuracy of GNNGuard in different GNN tasks? **(RQ2)**: How is the robustness and uniqueness of GNNGuard? **(RQ3)**: How do different settings impact GNNGuard' effectiveness?

### 4.1 Experiment Setup

*4.1.1 Tasks and Datasets.* We consider two graph-level tasks, graph classification [40] and graph matching [2], one node-level task, node classification [13] and one edge-level task, link prediction [41]. The overview of the dataset of each task is provided in Table A.2. **EN-ZYMES** is a protein graph dataset for 6-class classification task [4]; **PROTEINS** is a protein graph dataset, where nodes represent the amino acids[8]; **AIDS** is a collection of chemical compounds from antivirus screens and has been used in several existing works on graph similarity search [2]; **LINUX** consists of program dependency graphs (PDGs) generated from the Linux kernel[2, 35]; **Cora** and **CiteSeer** are single graphs in which the nodes and edges correspond to documents and citation links[1]. Note that for each data set, we split the dataset into train/validation/test by the ratio of 7/1/2. The training set is used to train the target GNN, the validation set is used to determine hyper-parameters, and the test set is used to measure the performance of the target GNN. If node attributes are missing in the dataset, we assign random values to all nodes and the graph fingerprint. These attributes remain fixed during the optimization of the graph fingerprint.

*4.1.2 GNN Preparation.* The GNN architectures used for different graph tasks are presented in Appendix. Here we detail the training strategies of these GNNs and the strategies of post-process. For each

type target GNN, we construct positive GNNs and negative GNNs for learning. Following the existing fingerprinting framework [5, 33], we apply the mentioned obfuscation techniques with different configurations on the target GNNs to derive the positive models: (1) **Fine-tuning & Partial Retraining.** We consider four types of fine-tuning and partial retraining, i.e., fine-tuning/partial retraining the last layer or all layers of GNNs, and we set the number of epochs of both fine-tuning and retraining as 10. (2) **Graph distillation.** For each target GNN we select all other GNNs having different architectures as the student models. We follow the graph distillation technique in study [7], and use the outputs of the last layers to implement distillation. As for the data used to query the teacher model, we sample 50% ∼ 80% subgraph from the original graph to query the outputs of the teacher model. To construct negative models, we use different random seeds to initialize different GNNs of different architectures and train them from scratch on the training dataset. For each task, we construct 200 positive/negative GNNs; we split them randomly by 1: 1 into the training and test set.

*4.1.3 Baselines.* We compare GNNGuard with two fingerprinting baselines for conventional deep neural networks, which are Deep-Fool [33] and IPGuard [5]. We modify their fingerprint construction to the graph fingerprint optimization strategies. Note that these two methods are only applicable to classification tasks, i.e., node classification and graph classification.[4]

*4.1.4 Evaluation Metrics.* (1) Robustness - measures the proportion of positive suspect model being accurately recognized, i.e. *True positive*; (2) Uniqueness - measures the proportion of negative suspect model being accurately recognized, i.e., *True negative* [5]. (3)

---

[4]we also compare the proposed fingerprinting framework with watermarking approaches to protect the IP of GNNs, which is presented in the Appendix.

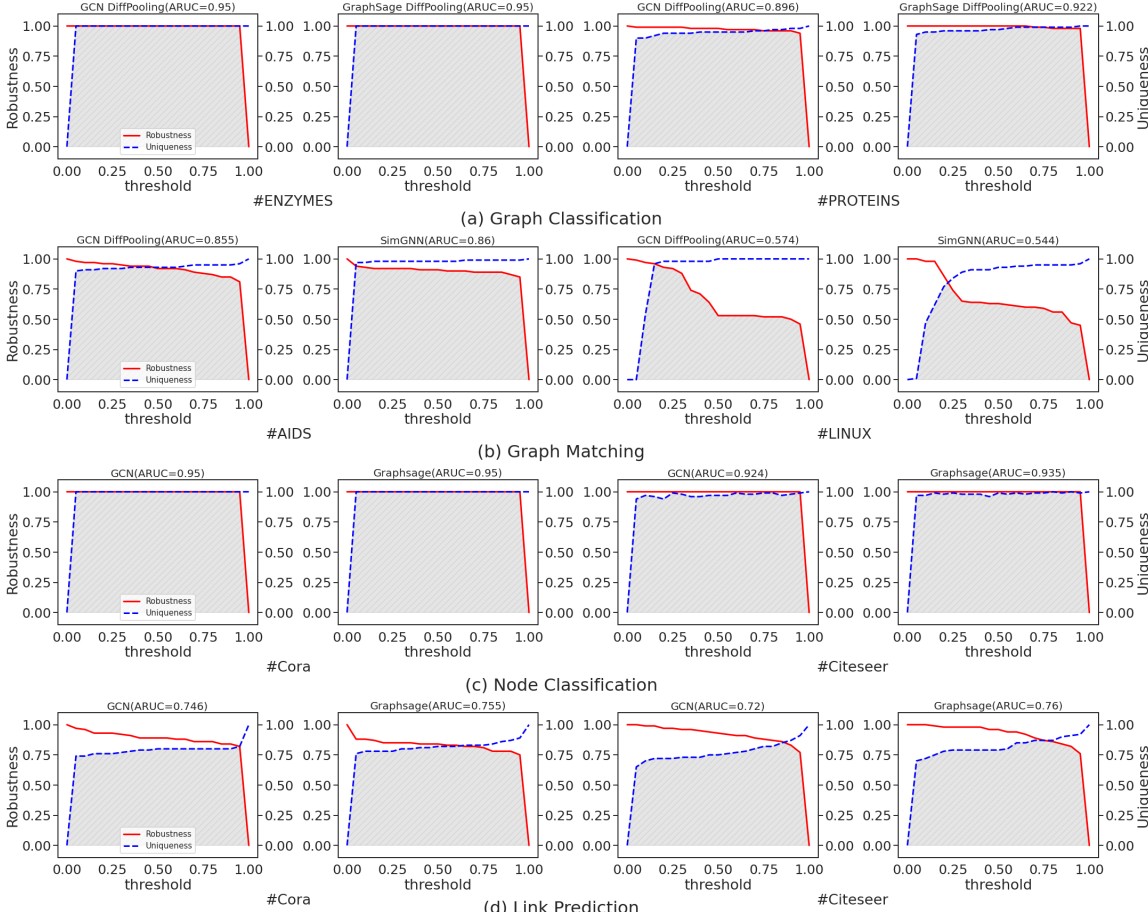

**Figure 2: Curves of robustness and uniqueness of different GNN tasks, GNN architectures, and datasets, where the ARUC is reported in the figure title. Update both $X$ and $A$.**

ARUC - Area under the Robustness-Uniqueness Curves (ARUC) - measures the area of the intersection region under the robustness and uniqueness curve when the threshold varies in $(0, 1)$. (4) Mean Test Accuracy - The fraction of accurately recognized positive models and negative models, and is calculated by $\frac{\#TP+\#TN}{\#Suspect\ models}$. It is reported on the average of the different thresholds.

*4.1.5 Other Settings.* With no further specifications, we set the size of the fingerprints of our framework and two baselines as $N = 64$, and the number of nodes as $n = 32$. For all GNNs, we set the depth of the neighborhood aggregation mechanism as 3 to avoid over-smoothing or under-smoothing [15]. The learning rate is set as 0.001 and the number of iterations as 1000. The Univerifier is implemented as a three-layer fully-connected neural network with the Leaky-ReLU and the hidden layer size list is [128, 64, 32]. We use the PyTorch Geometric[5] library to implement all models.

## 4.2 RQ1: Overall Comparison

We compare the effectiveness of GNNGuard with two baselines as shown in Table 1. As we can see, the mean test accuracy of GNNGuard is up to 1.00 on the ENZYMES dataset of the graph

[5]https://github.com/rusty1s/pytorch_geometric

classification task, and on the Cora dataset of the node classification task. This means that in $\tau \in (0, 1)$, GNNGuard can recognize 100% of positive models and negative models. On other datasets and tasks, the lowest mean test accuracy is 75.2% of the Cora dataset of Link prediction, which means that in the worst case, the proposed fingerprinting framework can accurately recognize more than 70% of positive models and negative models. It is similar to the performance of fingerprinting traditional DNN models. As for two baselines, the best accuracy of DeepFool is 95.7% but is often lower than 70% or even 50%. IPGuard aims at characterizing the decision boundary of target GNNs, which is not applicable to tasks like matching and link prediction. Nevertheless, the largest accuracy of IPGuard is lower than 60%, and is around 50%. This demonstrates that this verification rule is not efficient in fingerprinting GNNs. The performance of DeepFool and IPGuard demonstrates the advantage of introducing a learnable verifier.

Among four different GNN tasks, the verification accuracy of graph classification and node classification is better than graph matching and link prediction. We infer the main reason is that graph matching and link prediction tasks are more complicated than graph classification and node classification [2, 13]. Therefore, the target model's behaviors are more difficult to characterize. Besides, we also infer that this phenomenon is related to the performance of

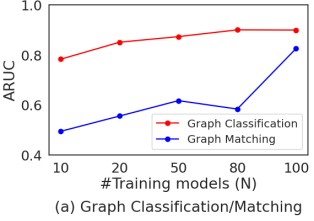

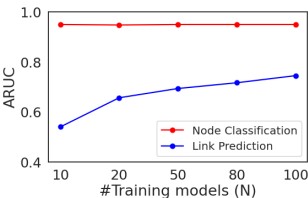

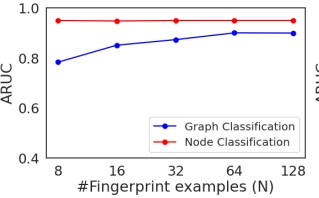

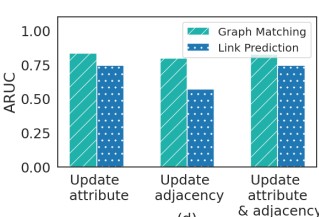

(a) Graph Classification/Matching   (b) Node Classificaiton/Link Prediciton   (c) Graph/Node Classification   (d)

Figure 3: (a)-(c) The number of fingerprint examples/training models vs. ARUC. (d) Different settings of graph fingerprint construction vs. ARUC. The performance of different settings is represented as "Update both"≈ "Update X">"Update A".

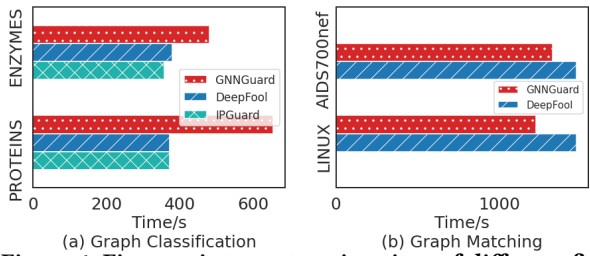

(a) Graph Classification   (b) Graph Matching

Figure 4: Fingerprint construction time of different fingerprinting frameworks.

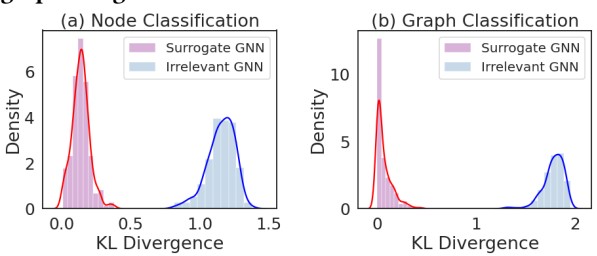

Figure 5: KL-divergence of the surrogate (irrelevant) GNNs and the target GNN calculated by Univerifier.

the target GNN itself. GNNs perform better on graph classification and node classification than graph matching and link prediction, which makes the graph construction module utilize more accurate information about the GNNs to construct fingerprints. As for different datasets, dataset like LINUX has more labels, and thus of which the verification accuracy is lower.

## 4.3 RQ2: Robustness & Uniqueness

Except for mean test accuracy, it is important to verify the robustness, uniqueness, and ARUC, shown in Fig 2. We can see from the figure, the robustness and uniqueness of GNNGuard are promised, and the ARUCs are up to 0.95. The metrics are better on graph classification and node classification (e.g., ARUC is up to 0.95) than graph matching and link prediction. When verifying on the LINUX dataset of graph matching, the lowest ARUC is 0.544. Besides, the model is not good at recognizing positive models in this case. We infer the main reason is that on the LINUX dataset there are 1m labels which makes the model's outputs extremely complicated. Then a slight model modification can largely influence the outputs of such models. Therefore, it is more difficult to verify such models by models' outputs. One efficient way to alleviate this is to utilize an additional validation set to choose an appropriate threshold to balance the uniqueness and robustness [5]. Robustness and uniqueness curves of DeepFool and IPGuard are shown in Appendix (A.4).

## 4.4 RQ3: Study of GNNGuard

### 4.4.1 Hyper-parameter Sensitivity.
First, we study different settings of graph fingerprint generation, i.e., only updating the adjacency matrix or only updating the node attribute matrix (shown in Fig. 3). Updating both of them can bring the best performance while updating the adjacency matrix can also achieve significant performance. Second, we study how important settings influence verification performance, in Fig 3 (a)-(b). With more ensembles composed of positive and negative GNNs, the ARUC is higher. But training these models needs computational resources thus it is better to choose an appropriate amount by balancing the budget and performance. Third, in Fig 3 (c), in graph classification, increasing the number of fingerprint examples will not significantly influence the verification performance; while in node classification, #Fingerprints = 64 achieves the best ARUC.

### 4.4.2 Time Complexity.
Furthermore, we empirically study the fingerprint construction time complexity of our framework and baseline models. As Fig. 5 shows, GNNGuard is the either second or the first slow method. The root reason is the joint-learned mechanism, which takes more time to converge. On large datasets like Citeseer or LINUX, the running time of GNNGuard is shorter, but DeepFool takes a long time to converge especially on Citeseer.

### 4.4.3 Study of Univerifier.
First, we visualize the KL divergence between suspect GNNs and target GNN in Fig 5. As we can see, in the proposed framework, the surrogate GNNs are consistent with the target GNN while the irrelevant GNNs are largely different from the target GNN. Second, we aim to verify that the learned Univerifier can accurately recognize suspect GNNs that it does not see during the training stage, the results are shown in Table A.5 and Table A.4.

## 5 CONCLUSION

In this paper, we present GNNGuard, the first GNN fingerprinting framework which is able to construct fingerprints in the form of graphs and be applicable to GNNs for multiple downstream tasks. Extensive experiments validate that GNNGuard is effective in protecting GNNs from model stealing, and bring noticeable improvement over several state-of-the-art IP protection methods. Besides, the normal utility of the model will not be influenced. In future work, we are going to validate and evaluate GNNGuard on other typical tasks such as graph clustering. Moreover, it would be meaningful to deploy and validate GNNGuard on real-world platforms to defend against model piracy.

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

# A  TECHNICAL APPENDIX

## A.1  More Descriptions

The following are more descriptions of datasets Tabel A.2, model extraction strategies and post-processing techniques.

### Table A.2: Statistics of datasets.

|            | #Graphs /Nodes | #pairs/labels | Graph Meaning      |
| ---------- | -------------- | ------------- | ------------------ |
| ENZYMES    | 600            | 6             | Macromolecules     |
| PROTEINS   | 1113           | 2             | Bioinformatics     |
| AIDS       | 700            | 490k          | Chemical Compounds |
| LINUX      | 1000           | 1m            | Program Dependence |
| Cora       | 2708           | 7             | Documents Citation |
| Citeseer   | 3327           | 6             | Documents Citation |

To imitate the behavior of attackers in the wild, we consider different model extraction attacks and other post-processing techniques, i.e., model extraction by distillation [7], fine-tuning [5], and pruning [14], and regard these models as positive models. For fine-tuning, we fine-tune the target GNN on a leaved-out dataset of the original dataset; for pruning, we set a certain number of parameters with the smallest absolute values to zero; as for distillation, we utilize different GNN architectures as stolen models. The stolen models utilize similar training data to distill knowledge from the source GNN. The epoch of fine-tuning is set as 20; while the epoch of distillation is set as 1000. Note that the positive models used to achieve robust verification are different from the positive models used in the testing phase. For instance, we generate 400 positive models with the aforementioned strategies and split them by $1:1$ into training and test set.

For negative GNNs, we utilize the same dataset to train GNNs with different or the same architectures. Note that we also generate 400 negative models and randomly split them by $1:1$ into training and test set.

## A.2  Theoretical Proofs

We first present the theoretical analyses on node classification. Before introducing our theoretical guarantee of robustness, we first present Lemma 3.1 to describe the relationship between the outputs of target GNN and surrogate GNN.

Proof. Proof of Lemma 3.1.

Note that the proof is based on proof in [17]. Given the GNN model $f_w$, we use $H_j$ to represent the $j$-th layer output, and utilize $H'_j$ to represent the $j$-th layer output of surrogate GNN $f_{w+u}$. We also define these, $\Delta_j = H'_j - H_j$, $\Psi_j = \max_j | \Delta_j[i,:] |_2$, $\Phi_j = \max_j | H_j[i,:] |_2$, $u^*_j = \arg\max | \Delta_j[i,:] |_2$, and $v^*_j = \arg\max | H_j[i,:] |_2$.

**Derive the upper bound of node representations.** For any layer $j$, we are going to derive the upper bound of the $\ell_2$ norm of node

representation as follows,

$$\Phi_j = \max_j | H_j[i,:] |_2 = \max_i | \sigma_j(\tilde{L}H_{j-1}W_j)[i,:] |_2 = | \sigma_j(\tilde{L}H_{j-1}W_j)[v^*_j,:] |_2 \tag{5}$$

$$\leq | (\tilde{L}H_{j-1}W_j)[v^*_j,:] |_2 \quad \text{(Lipschitz property of ReLU under vector 2-norm)} \tag{6}$$

$$= | (\tilde{L}H_{j-1})[v^*_j,:]W_j |_2 \leq | (\tilde{L}H_{j-1})[v^*_j,:] |_2 \|W_j\|_2 \tag{7}$$

$$\leq | (\sum_{k \in N_{v^*_j}} \tilde{L}[v^*_j,k]H_{j-1}[k,:]) |_2 \|W_j\|_2 \tag{8}$$

$$\leq \sum_{k \in N_{v^*_j}} \tilde{L}[v^*_j,k]\Phi_{j-1}\|W_j\|_2 \leq d^{1/2}\Phi_{j-1}\|W_j\|_2 \tag{9}$$

$$\leq d^{j/2}B \prod_{i=1}^{j} \|W_i\|_2 \text{ (Unroll to } \Phi_0 = B.) \tag{10}$$

**Derive the upper bound on the change of node representations after perturbations.** For any layer $j$ (except the output layer), we derive the difference between the target GNN and surrogate GNN by bounding the change of their node representations' $\ell_2$ norm as follows.

$$\Psi_j = \max_i | H'_j - H_j |_2 = | \sigma_j(\tilde{L}H'_{j-1}(W_j + U_j))[u^*_j,:] - \sigma_j(\tilde{L}H_{j-1}W_j)[u^*_j,:] |_2 \tag{11}$$

$$\leq | (\tilde{L}H'_{j-1}(W_j + U_j))[u^*_j,:] - (\tilde{L}H_{j-1}W_j)[u^*_j,:] |_2 \tag{12}$$

$$= | (\tilde{L}H'_{j-1})[u^*_j,:](W_j + U_j) - (\tilde{L}H_{j-1})[u^*_j,:]W_j |_2 \tag{13}$$

$$\leq | \tilde{L}H'_{j-1}[u^*_j,:] - \tilde{L}H_{j-1}[u^*_j,:] |_2 \|W_j + U_j\|_2 + | (\tilde{L}H_{j-1})[u^*_j,:] |_2 \|U_j\|_2 \tag{14}$$

$$\leq d^{1/2}\Psi_{j-1}\|W_j + U_j\|_2 + d^{1/2}\Phi_{j-1}\|U_j\|_2. \tag{15}$$

Given $d^{1/2}\Psi_{j-1}\|W_j+U_j\|_2 + d^{1/2}\Phi_{j-1}\|U_j\|_2$, we can unroll them to $\Psi_0$ and $\Phi_0$. As $\Psi_0 = 0$, then we have,

$$\Psi_j \leq \sum_{k=0}^{j} -1 d^{\frac{j-k}{2}} \Phi_k \|U_{k+1}\|_2 (\prod_{i=k+2}^{j} \|W_i + U_i\|_2) \tag{16}$$

$$\leq \sum_{k=0}^{j-1} d^{\frac{j-k}{2}} (d^{k/2}B \prod_{i=1}^{k} \|W_i\|_2)\|U_{k+1}\|_2 (\prod_{i=k+2}^{j} \|W_i + U_i\|_2) \tag{17}$$

$$\leq \sum_{k=0}^{j-1} d^{j/2}B \prod_{i=1}^{j} \|W_i\|_2 \sum_{k=1}^{j} \frac{\|U_k\|_2}{\|W_k\|_2} (1 + \frac{1}{\ell})^{j-k} \tag{18}$$

**Derive the upper bound on change of output layer after perturbations.** After this, we can derive the upper bound for the output layer. Specifically, for different graph tasks, the upper bounds of output layers are different. For node classification, the upper bound is the maximum change of a node representation. We first derive the upper bound for node classification as follows.

$$| \Delta_\ell |_2 \leq | \Delta_\ell[u^*_\ell,:] |_2 = | H'_{\ell-1}(W_\ell + U_\ell) - H_{\ell-1}W_\ell |_2 \tag{19}$$

$$\leq \|W_\ell + U_\ell\|_2 | \Delta_{\ell-1} | + \|U_\ell\|_2 | H_{\ell-1} |_2, \tag{20}$$

$$\leq eBd^{\frac{\ell-1}{2}} (\prod_{i=1}^{\ell} \|W_i\|_2)[\sum_{k=1}^{\ell} \frac{\|U_k\|_2}{\|W_k\|_2}] \tag{21}$$

End the proof.  □

**Table A.3: Clean performance of different graph-based tasks, including target GNN, negative GNNs and positive GNNs.**

|  | Node Classification | Link Prediction | Graph Classification | Graph Matching (P@20) |
|---|---|---|---|---|
| Target GNN | 0.7525 | 0.908 | 0.600 | 0.331 |
| Positive GNNs | 0.731 | 0.900 | 0.565 | 0.342 |
| Negative GNNs | 0.735 | 0.900 | 0.557 | 0.341 |

PROOF. Proof of Theorem 3.2. The purpose of the verification is to determine whether the difference between the output distributions of a suspect GNN and the source GNN is larger than threshold $\lambda$, which can be formulated as follows.

$$P(\Delta_1 < \lambda) = P(x > \lambda \mid x \sim \mathcal{N}(\mu_0 - (\mu_0 + \mathcal{B}), \sigma^2)) \quad (22)$$

$$= P(x < \lambda \mid x \sim \mathcal{N}(-\mathcal{B}, \sigma^2)) = \Phi(\frac{\lambda + \mathcal{B}}{\sigma})$$

$$P(\Delta_2 < \lambda) = P(x > \lambda \mid x \sim \mathcal{N}(\mu_0 - (\mu_0 - \mathcal{B}), \sigma^2)) \quad (23)$$

$$= P(x < \lambda \mid x \sim \mathcal{N}(\mathcal{B}, \sigma^2)) = \Phi(\frac{\lambda - \mathcal{B}}{\sigma}).$$

Note that $\Delta = \|f_w - f_{w+u}\|$. Therefore, the total probability can be derived as follows.

$$P(\text{Diff} < \lambda) = P(\Delta_1 < \lambda) + P(\Delta_2 < \lambda) = \Phi(\frac{\lambda - \mathcal{B}}{\sigma}) + \Phi(\frac{\lambda + \mathcal{B}}{\sigma}). \quad (24)$$

End the proof. □

After this, we present a theoretical analysis of the robustness guarantee of the graph classification task. The difference between node classification and graph classification is the derivation of the upper bound on GNN outputs, i.e., $\mathcal{B}_{gcls}$. Then after this, the bound can be used in Theorem 3.2 to derive the robustness guarantee of graph classification.

LEMMA A.1. *Suppose that a GNN $f_w$, of which the parameters are denoted as $\{W_i\}_{i=1}^{\ell}$. Pertubations $\{U_i\}_{i=1}^{\ell}$ are added on $f_w$ to obtain its surrogate GNN $f_{w+u}$ (positive GNN). The differences of graph classification task in outputs of $f_w$ and $f_{w+u}$ given the same input graph can be bounded as follows,*

$$\mid f_{w+u}(X, A) - f_w(X, A) \mid \leq eB(n)d_{max}^{\frac{\ell-1}{2}}(\prod_{i=1}^{\ell} \|W_i\|_2) \sum_{k=1}^{\ell} \frac{\|U_k\|_2}{\|W_k\|_2} = \mathcal{B}_{gcls}. \quad (25)$$

*Note that $d_{max}$ is the largest node degree on the input graph.*

PROOF. Proof of Lemma A.1.

For graph classification, the upper bound is the maximum change of a graph representation, i.e., the pooling results of node representations. Therefore, we can utilize the results in Eq 21 to derive the output bound for graph classification with different pooling operations, which can be formulated as follows.

$$\mid \Delta_\ell \mid_2 \leq \mid \Delta_\ell[u_\ell^*, :] \mid_2 = \mid \frac{1}{n}1_n H'_{\ell-1}(W_\ell + U_\ell) - \frac{1}{n}1_n H_{\ell-1}W_\ell \mid_2, \text{Mean pooling} \quad (26)$$

$$\leq \|W_\ell + U_\ell\|_2 \mid \Delta_{\ell-1} \mid + \|U_\ell\|_2 \mid H_{\ell-1} \mid_2, \quad (27)$$

$$\leq eBd^{\frac{\ell-1}{2}}(\prod_{i=1}^{\ell} \|W_i\|_2)[\sum_{k=1}^{\ell} \frac{\|U_k\|_2}{\|W_k\|_2}] \quad (28)$$

$$\mid \Delta_\ell \mid_2 \leq \mid \Delta_\ell[u_\ell^*, :] \mid_2 = \mid 1_n H'_{\ell-1}(W_\ell + U_\ell) - 1_n H_{\ell-1}W_\ell \mid_2, \text{Sum pooling} \quad (29)$$

$$\leq \|W_\ell + U_\ell\|_2 \mid \Delta_{\ell-1} \mid + \|U_\ell\|_2 \mid H_{\ell-1} \mid_2, \quad (30)$$

$$\leq eBnd^{\frac{\ell-1}{2}}(\prod_{i=1}^{\ell} \|W_i\|_2)[\sum_{k=1}^{\ell} \frac{\|U_k\|_2}{\|W_k\|_2}] \quad (31)$$

$$\mid \Delta_\ell \mid_2 \leq \mid \Delta_\ell[u_\ell^*, :] \mid_2 = \mid H'_{\ell-1}[u_j^*, :](W_\ell + U_\ell) - H_{\ell-1}[u_j^*, :]W_\ell \mid_2, \text{Max pooling} \quad (32)$$

$$\leq \|W_\ell + U_\ell\|_2 \mid \Delta_{\ell-1} \mid + \|U_\ell\|_2 \mid H_{\ell-1} \mid_2, \quad (33)$$

$$\leq eBd^{\frac{\ell-1}{2}}(\prod_{i=1}^{\ell} \|W_i\|_2)[\sum_{k=1}^{\ell} \frac{\|U_k\|_2}{\|W_k\|_2}] \quad (34)$$

End the proof. □

---

**Algorithm 1** Algorithm of GraphFingers.

---

1: **Input:** Trained target model $f$, positive model set $\mathcal{F}_+$, negative model set $\mathcal{F}_-$, epochs $e_1, e_2$, learning rate $\alpha, \beta$, initialized fingerprints $\mathcal{I}$, initialized FPVerifier $\mathcal{V}_\omega$, update signal flag.
2: **Return:** Fingerprint $\mathcal{I} = \{I^p\}_{p=1}^{P}$, FPVerifier $\mathcal{V}_\omega$.
3: **while** not converge **do**
4:     $\mathcal{L} = 0$
5:     **for** $\hat{f} \in \{f\} \cup \mathcal{F}_+ \cup \mathcal{F}_-$ **do**
6:         if $\hat{f} \in \{F\} \cup \mathcal{F}_+$:
7:         $\mathcal{L} += \log(\mathcal{V}_\omega(\text{Concat}(\{\hat{f}(I^p)\}_{p=1}^{P})))$,
8:         else:
9:         $\mathcal{L} += \log(1 - \mathcal{V}_\omega(\text{Concat}(\{\hat{f}(I^p)\}_{p=1}^{P})))$.
10:     **end for**
11:     **if** flag == 0 **then**
12:         **for** $e = 0, ..., e_1$ **do**
13:           Update $I^p \in \mathcal{I}$ according to strategies in Section 3.1.
14:         **end for**
15:         flag = 1.
16:     **else**
17:         **for** $e = 0, ..., e_2$ **do**
18:           Update $\omega$ by $\beta \nabla_\omega \mathcal{L}$.
19:         **end for**
20:         flag = 0.
21:     **end if**
22: **end while**

---

## A.3 Algorithms of implementing GraphFingers on different tasks

We present the overall algorithm for GraphFingers in Algorithm 1. Besides, we present different implementations of GraphFingers on different tasks. The graph construction of GNN for the graph classification task is shown in Algorithm 2; while graph matching is shown in Algorithm 3.

---

**Algorithm 2** Graph fingerprint construction of GNN for a graph classification task.

---

1: **Input:** Loss $\mathcal{L}$, graph fingerprint $\mathcal{I}_t$, learning rate $\alpha$.
2: **Return:** Updated graph fingerprint $\mathcal{I}_{t+1}$.
3: **for** $I^p \in \mathcal{I}_t$ **do**
4: $\quad (X_t^p, A_t^p) \leftarrow I^p$
5: $\quad X_{t+1}^i \leftarrow X_t^i + \alpha \nabla_{X^i} \mathcal{L}$
6: $\quad A_{t+1}^i \leftarrow \text{Flip}\big(A_t^i, \text{Rank}(\nabla_{A^i} \mathcal{L})\big)$
7: **end for**

---

**Algorithm 3** Graph fingerprint construction of GNN for a graph matching task.

---

1: **Input:** Loss $\mathcal{L}$, graph fingerprint $\mathcal{I}_t$, learning rate $\alpha$.
2: **Return:** new graph fingerprint $\mathcal{I}_t$.
3: **for** $I^p \in \mathcal{I}_t$ **do**
4: $\quad$ **for** $G^{i,p} \in I^p$ **do**
5: $\quad\quad X_{t+1}^{i,p} \leftarrow X_t^{i,p} + \alpha \nabla_{X^{i,p}} \mathcal{L}$
6: $\quad\quad A_{t+1}^{i,p} \leftarrow \text{Flip}\big(A_t^{i,p}, \text{Rank}(\nabla_{A^{i,p}} \mathcal{L})\big)$
7: $\quad$ **end for**
8: **end for**

---

**Algorithm 4** Graph fingerprint construction of GNN for a node classification task/Link prediction task.

---

1: **Input:** Loss $\mathcal{L}$, graph fingerprint $\mathcal{I}_t$, learning rate $\alpha$.
2: **Return:** new graph fingerprint $\mathcal{I}_t$.
3: $X_{t+1} \leftarrow X_t + \alpha \nabla_X \mathcal{L}$
4: $A_{t+1} \leftarrow \text{Flip}\big(A_t, \text{Rank}(\nabla_A \mathcal{L})\big)$

---

## A.4 More Evaluation Results

First, more results of different settings on the switch are shown in Fig. A.6, Fig. A.7, Fig. A.8, Fig. A.9, Fig. A.10 and Fig. A.11. In summary, the verification performance is 'Update A+X>Update X>Update A'. Note that different from graph classification, the input of the graph matching task is a pair of graphs. If the number of the fingerprint is 64, there are 64 graphs used as fingerprint examples for the graph classification task; while there are 32 pairs of graphs used as the fingerprint examples for the graph matching task.

*A.4.1 Different architectures of the dynamic verification mechanism.* Besides, we instantiate different architectures of the dynamic verification mechanism, i.e., Logistic regression and Convolutional neural network. As we can see from Fig. A.12, DNN classifier achieves the best performance. We postulate that the main reason is that DNN classifier is more accurate in classifying continuous features.

*A.4.2 Results of GNN watermarking.* To verify the effectiveness of GNN watermarking, we also validate the GNN watermarking [45] on the graph classification dataset. This method uses a set of random graphs and pre-defined labels as watermarks and embeds the watermark into the source GNN by fine-tuning it with a new regularization loss and a clean training loss. In Fig. A.13, we can see that GNNGuard can achieve much better performance than GNN watermarking. The robustness of GNN watermarking is poor because the random graphs are not as transferable as the adversarial examples of the image domain. Furthermore, the clean performance of GNN after watermarking drops by $3\% - 5\%$, i.e., Fig. A.14. The *ACC* is measured on a clean test set by the original source model and the watermarking source model. The accuracy loss is less than the accuracy loss reported in the original paper because we utilize additional training data to maintain the normal utility. GNNGuard can achieve 100% true negative and true positive without incurring accuracy loss.

*A.4.3 Generalization of GNNGuard on Different Positive Models.* We use different architectures for distillation on the training and test set, i.e., for the training model set, we utilize GIN and GCN to distill the source GNN, while for the test model set, we use GraphSage and GAT to distill the source GNN. Besides, we leave out existing distilled GNNs from the training model set, i.e., in the training model set, there are only post-processing techniques including finetuning and pruning. Finally, we test our fingerprints framework on a new distillation technique, i.e., data-free model distillation [7]. All these results are shown in Table A.4.

Besides, we also test the proposed fingerprinting framework on verifying GNN model with new pooling techniques, e.g., eigenpooling [21], and new graph-based task, e.g., graph generative tasks [23]. The results are shown in Table A.5.

*A.4.4 More Time Complexity Analyses of GNNGuard.* We present time complexity analyses on the number of edges on graph fingerprint in Fig A.15. The number of edges is set as the ratio of the original graph fingerprint. On different levels of graph-based tasks, the mean test accuracy is higher than 85% or 95%, demonstrating that even with small graph fingerprint, effectiveness can still be promised.

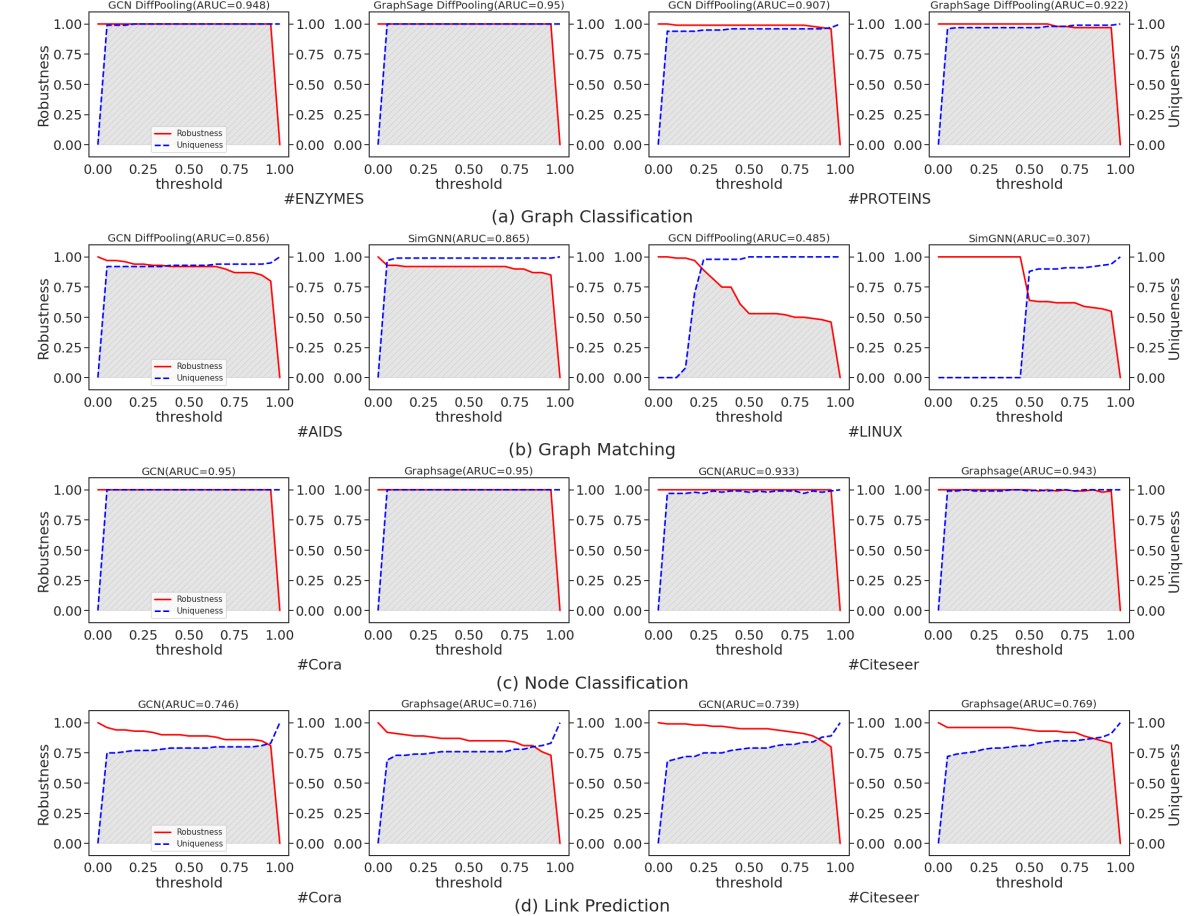

**Figure A.6: Curves of robustness and uniqueness of different GNN tasks, GNN architectures and datasets of GNNGuard, where the ARUC is reported in the figure title. Update only the attribute matrix.**

**Table A.4: Generalization of GNNGuard on unseen architectures and post-processing operations.**

|  | Graph classification (Proteins) | Node classification (Cora) |
|---|---|---|
| New architectures | 0.93 | 1.00 |
| Leave-out post-processing | 0.83 | 1.00 |
| New post-processing | 0.97 | 1.00 |

**Table A.5: Verification effectiveness on new architectures and new tasks.**

|  | ARUC | Robustness | Uniqueness | Accuracy |
|---|---|---|---|---|
| JKNet+eigenpooling | 0.93 | 1.00 | 0.98 | 0.94 |
| GAT+eigenpooling | 0.88 | 1.00 | 0.92 | 0.90 |
| Cora (generative model) | 0.83 | 1.00 | 0.87 | 0.90 |
| Citeseer (generative model) | 0.87 | 1.00 | 0.89 | 0.95 |

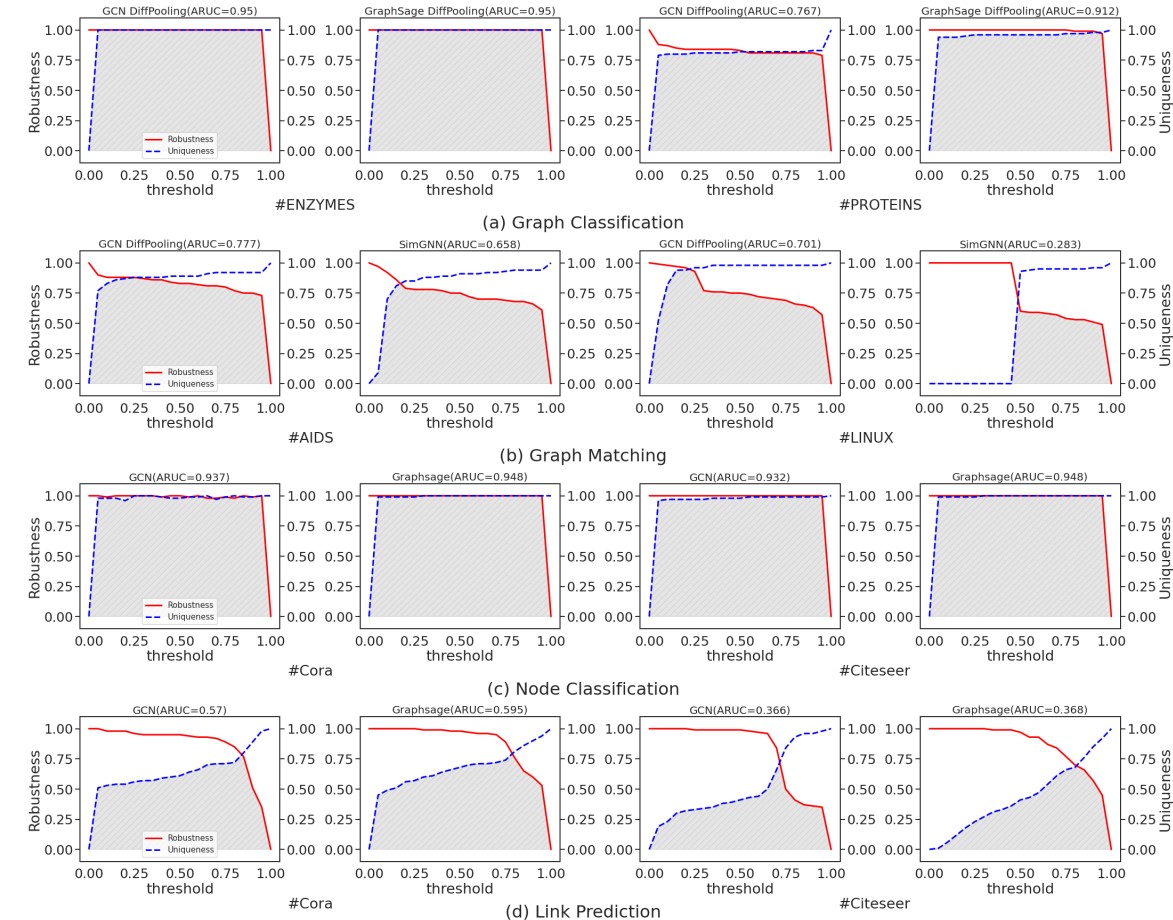

**Figure A.7: Curves of robustness and uniqueness of different GNN tasks, GNN architectures and datasets of GNNGuard, where the ARUC is reported in the figure title. Update only the adjacency matrix.**

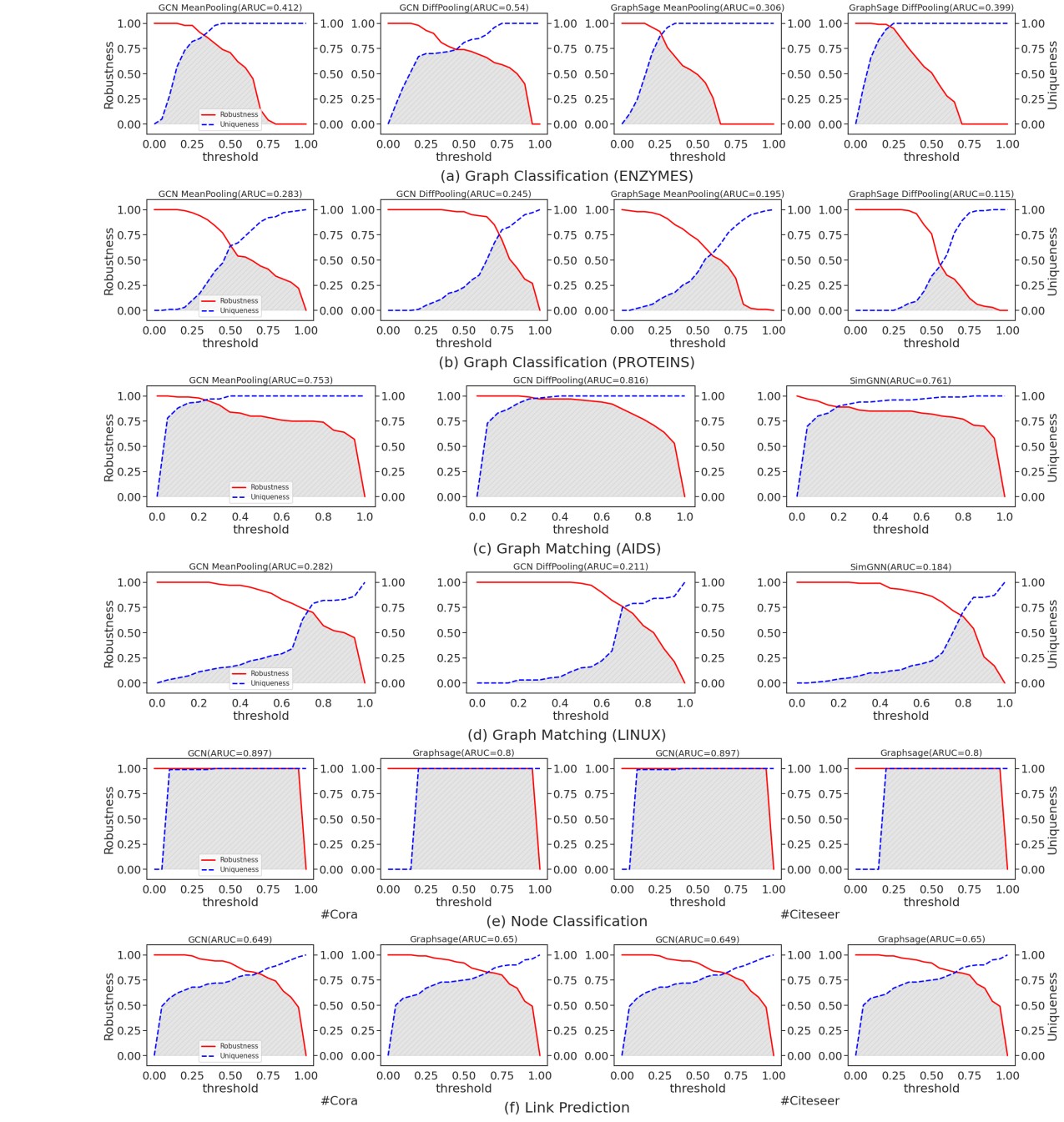

Figure A.8: Curves of robustness and uniqueness of different GNN tasks, GNN architectures and datasets of DeepFool, where the ARUC is reported in the figure title. Update only the attribute matrix.

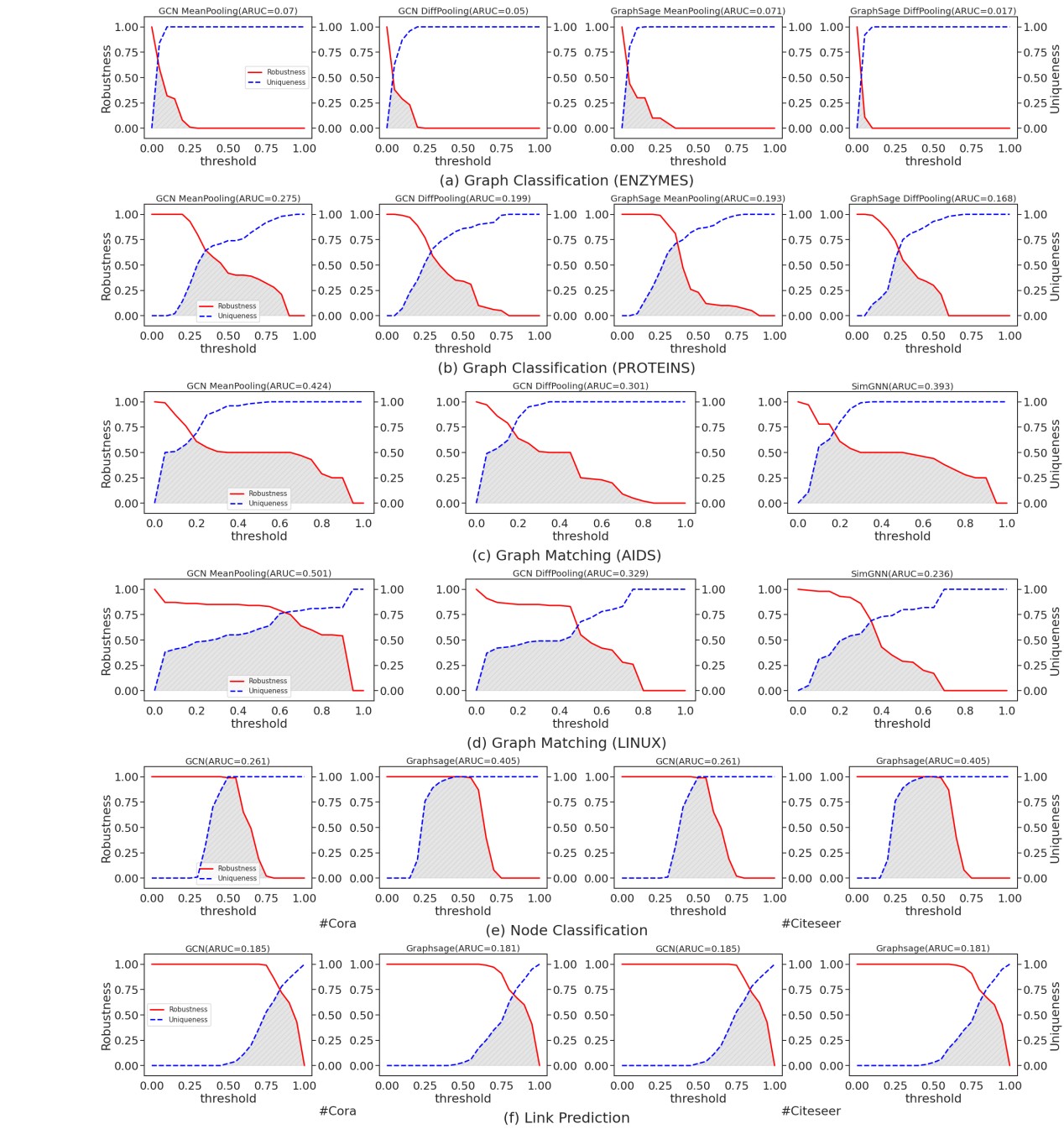

Figure A.9: Curves of robustness and uniqueness of different GNN tasks, GNN architectures and datasets of DeepFool, where the ARUC is reported in the figure title. Update only the adjacency matrix.

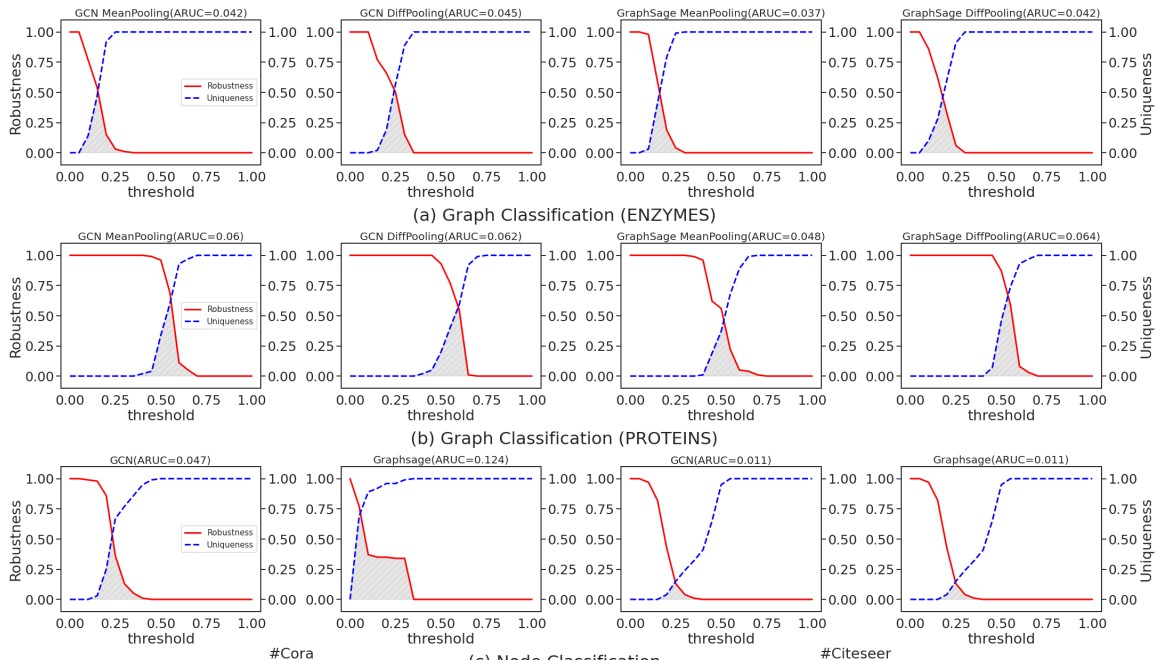

Figure A.10: Curves of robustness and uniqueness of different GNN tasks, GNN architectures and datasets of IPGuard, where the ARUC is reported in the figure title. Update only the attribute matrix.

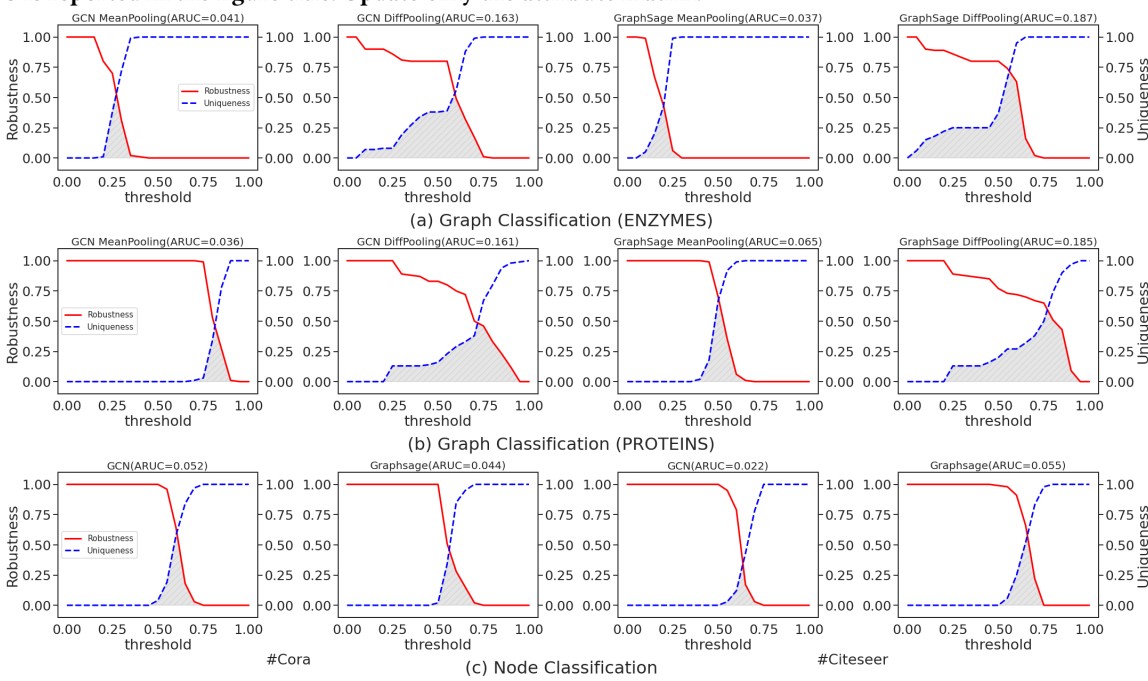

Figure A.11: Curves of robustness and uniqueness of different GNN tasks, GNN architectures and datasets of IPGuard, where the ARUC is reported in the figure title. Update only the adjacency matrix.

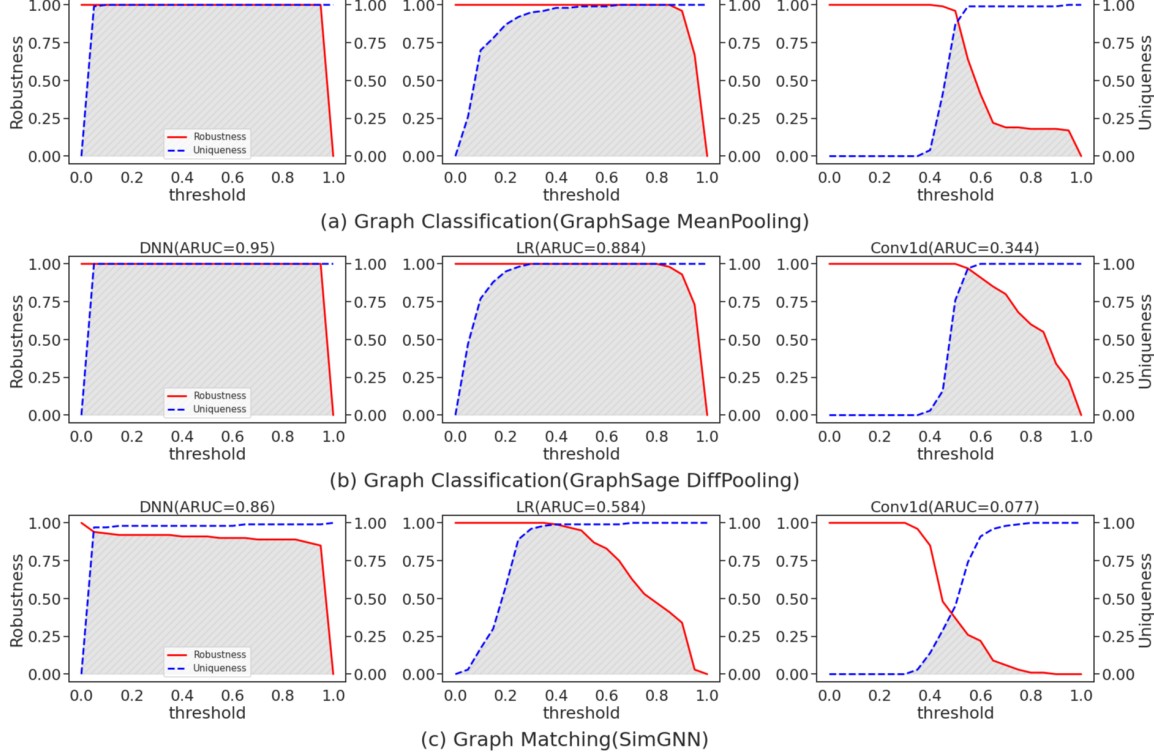

Figure A.12: Different instantiations of the Univerifier, i.e., Logistic regression and Convolutional neural network (Conv1d).

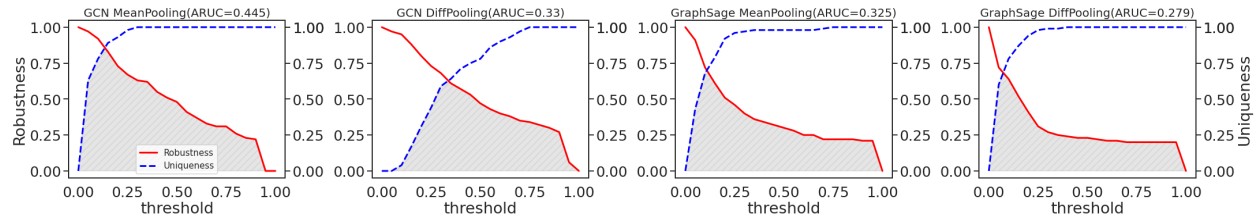

Figure A.13: Curves of robustness and uniqueness of GNN watermarking on ENZYMES of the graph classification task, where the ARUC is reported in the figure title. Update both $X$ and $A$.

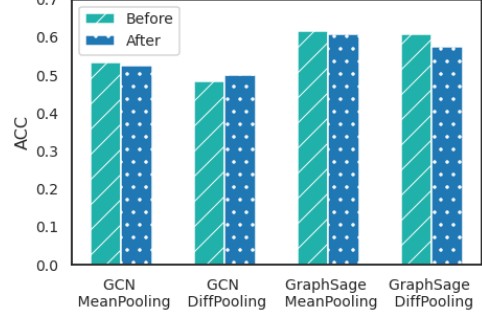

Figure A.14: Classification accuracy of the source GNN before/after model watermarking.

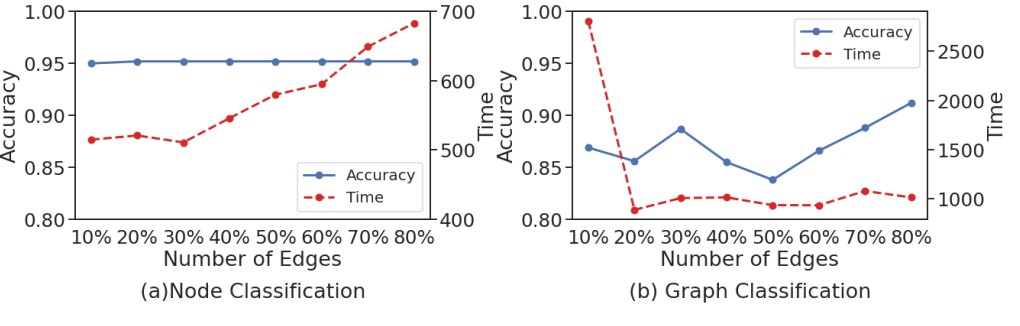

Figure A.15: Verification accuracy and Time complexity vs. Size of graph fingerprint.

