# OpenReview forum: "GNNGuard: A Fingerprinting Framework for Verifying Ownerships of Graph Neural Networks"
_ACM.org/TheWebConf/2024/Conference — TheWebConf24_

### Official Review · Reviewer_XHGU · 2023-11-22

**Novelty:** 5
**Technical Quality:** 5

**Review:**

This paper tailors the model fingerprinting for GNNs and propose a unified framework of fingerprinting GNNs which is agnostic to multiple downstream tasks with Graph Fingerprint and a Unified Verification Mechanism. Without modifications of the model parameters, this method achieves excellent performance on both graph-classification and node classification task.

Pros:
1. The paper is well written easy to understand.
2. The propose framework achieves high accuracy on both graph-classification and node classification task.
4. The robustness and uniqueness of GNNGuard is promised.
5. Theoretical proofs are comprehensive.

Cons:
1. It is advised to show why model watermarking approaches don’t work on GNNs.
2. Latest baselines are needed. It is better to compare with advanced baselines based on fingerprint and watermarks to demonstrate its effectiveness.

**Questions:**

Could you kindly show the results why traditional watermarking don’t work and the advantages of model fingerprinting over watermarking approaches?

**Reviewer Confidence:**

4: The reviewer is certain that the evaluation is correct and very familiar with the relevant literature

**Scope:**

4: The work is relevant to the Web and to the track, and is of broad interest to the community

---

### Official Review · Reviewer_E2e4 · 2023-11-24

**Novelty:** 3
**Technical Quality:** 5

**Review:**

-Summary:

This paper proposed GNNGuard, a fingerprinting framework for GNNs that can determine whether a GNN model is a pirate model from a given model. Specifically, GNNGuard jointly trains a Univerifier and a graph fingerprint by distinguishing positive samples (pirated models) from negative samples (irrelevant models). Experimental results verify the efficacy of GNNGuard.

-Advantages:

(1) This paper is generally well-organized and easy to follow.

(2) GNNGuard achieved desirable performance on different graph-based tasks.

-Disadvantages:

(1) The motivation of GNNGuard is incomplete. The goal of GNNGuard is to identify the pirated GNNs, but there is no explicit definition of pirated GNNs. In real-world scenarios, one can train a GNN model with the same architecture on a similar dataset as the target GNN. Should we call such a GNN model a pirated model? The definition of pirated model can impact the application of GNN fingerprinting in the real world and some prior knowledge should be clarified if necessary. It would be better if the author could add some discussions on the pirated model to the paper.

(2) The novelty of GNNGuard is limited (or not fully presented). The author only mentioned that existing fingerprinting models only consider traditional DNNs without specifying the unique challenges of model fingerprinting on GNNs. It will be helpful if the authors can clarify the challenges of model fingerprinting on GNNs and how GNNGuard solves these challenges.

(3) The authors assume the output distribution of GNNs to be a Gaussian. If doing so, the error of true distribution and the given Gaussian is better to be considered in the theorem. At least, the error should be discussed empirically. In addition, positive GNNs share the same variance with the target GNN according to Theorem 3.2, which makes no sense to me.

(4) In the experiments, the authors assume the real-world pirated models to be either fine-tuned models or KD-based models (and mark them ground truth label 1). However, the real-world pirated models can be trained in an unexpected way. Hence, the transferability of GNNGuard can also be important. I suggest the authors add some experiments on this point.

(5) Background knowledge is over-presented. The authors introduced substantial background knowledge on model watermarking. However, GNNGuard is hardly relevant to the watermarking methods.

**Questions:**

Please see the disadvantages.

**Reviewer Confidence:**

2: The reviewer is willing to defend the evaluation, but it is likely that the reviewer did not understand parts of the paper

**Scope:**

4: The work is relevant to the Web and to the track, and is of broad interest to the community

---

### Official Review · Reviewer_B8Qn · 2023-11-24

**Novelty:** 5
**Technical Quality:** 5

**Review:**

This paper proposes a novel fingerprinting framework for GNN IP protection, which takes carefully generated data points and a verification model to distinguish the models stolen by piracy attack and irrelevant models. The authors addressed two GNN Obfuscation settings – partial retraining and distillation, as well as three popular GNN downstream tasks. To obtain the GNN fingerprint data and the Univerifier, this paper introduces a joint training framework, which learn parameters for both parts simultaneously. Extensive empirical experiments are conducted to evaluate the performance of the proposed framework.

**Questions:**

Strengths:

1)	This paper presents an interesting problem in the setting of GNN IP protection, and solves the problem with a novel data-centric approach. The authors thoroughly addressed three popular downstream tasks in graph ML, and discussed fingerprinting for all the three cases. The technical contribution of this paper is also satisfying.

2)	The empirical results of the proposed framework are convincing. In the experiments, the authors used a wide range of real-world datasets and addressed both fine-tuning and distillation in their GNN preparation, which ensures the soundness of the proposed framework.

3)	This paper is clearly written and most of the parts are easy to understand. The figures and the formulas are correct and precise. I am sure that the presentation quality of this paper is above the acceptance level.

Weaknesses:

1)	It seems that Theorem 3.2 only works for GCNs instead of a more general form of GNNs (e.g., MPNNs), but the authors use the words “GNNs” in subsection 3.5.2, which sounds a little bit exaggerative.

2)	The assumption in Theorem 3.2 looks too strong. It would be better if the authors could provide more evidence to explain why the output of trained GNNs follows a normal distribution. Is this a common assumption or are there any examples from real-world GNNs?

3)	In the experiments, the authors fixed the number nodes to n = 32. I think it would be better if the authors could provide an extra hyperparameter analysis on n and show the impact of n on the verification accuracy. This will improve the soundness of the proposed method, as well as the reliability of Theorem 3.2.

4)	Though the authors have provided detailed training algorithms in the appendix, the descriptions of the proposed method in Section 3.4 are not easy to follow, making this paper less self-contained.

5)	Some typos to be improved (e.g., GraphFingers -> GNNGuard).

**Ethics Review Description:**

N.A.

**Reviewer Confidence:**

3: The reviewer is confident but not certain that the evaluation is correct

**Scope:**

4: The work is relevant to the Web and to the track, and is of broad interest to the community

---

### Official Review · Reviewer_LdyK · 2023-11-24

**Novelty:** 4
**Technical Quality:** 4

**Review:**

The protection of intellectual property is an important issue in the field of machine learning, and GNNGuard provides an optimized solution to this problem. The framework is evaluated to be applicable to various GNN models and graph-related tasks, and the results indicate its effectiveness in protecting GNNs from model stealing. The authors use clear and concise language to describe their framework and experiments, and the figures and tables are well-designed and easy to understand. The originality of this work is moderate. While there have been previous works on GNN fingerprinting and IP protection, GNNGuard is claimed to be the first framework that can construct fingerprints in the form of graphs and be applicable to GNNs for multiple downstream tasks.

**Strength**

- The proposed framework is effective in protecting GNNs from model stealing.
- The experiments are thorough and well-designed, and the results are presented clearly and effectively.
- The joint learning approach for graph fingerprint and Univerifier is a contribution to the field.

**Weakness**
- The paper could offer a detailed discussion of the limitations and future directions of the proposed framework.

**Questions:**

Does the GNNGuard or any other fingerprinting and/or watermarking techniques have similar properties to the security property in cryptography? That is:
    1. Preimage Resistance: Given an output c, c is computationally infeasible to find m such that GNNGuard(m) = c;
    2. Second Preimage Resistance: Given a query m, m is computationally infeasible to find another query n ≠ m such that GNNGuard(m) = GNNGuard(n);
    3. Collision Resistance: It is computationally infeasible to find any query pair (m,n) such that GNNGuard(m) = GNNGuard(n).
The paper seems to have not discussed the probabilities of the occurrence of the above events. It might be more comprehensive if it added the above evaluations.
Besides, are there any potential privacy concerns with using GNNGuard, such as the possibility of leaking sensitive information about the target GNN during the fingerprinting process?

**Reviewer Confidence:**

2: The reviewer is willing to defend the evaluation, but it is likely that the reviewer did not understand parts of the paper

**Scope:**

4: The work is relevant to the Web and to the track, and is of broad interest to the community

---

### Official Review · Reviewer_AA9M · 2023-11-25

**Novelty:** 5
**Technical Quality:** 6

**Review:**

Summary:

The paper proposes a novel framework for protecting the intellectual property (IP) of graph neural networks (GNNs), which are models that can learn from graph-structured data. The framework, called GNNGuard, aims to prevent model piracy attacks that can steal the source GNN by querying with graph inputs. The framework uses a model fingerprinting technique to embed fingerprints in the form of graphs into the source GNN, and a learnable verifier to distinguish the fingerprinted GNN from other GNNs. The framework can be applied to any GNN model for various graph-related tasks. The paper evaluates the framework on various datasets and shows that it achieves effective and robust verification performance without degrading the normal utility of the GNN.


Strengths:

* The paper addresses a novel and important problem of protecting the IP of GNNs, which can enhance the security and reliability of GNNs in various applications.

* The authors introduce a novel and flexible framework that can embed fingerprints in the form of graphs into any GNN model, and use a learnable verifier to verify the ownership of the GNN without relying on any assumptions or prior knowledge. The framework can also handle different types of GNN models and paradigms.

* Extensive experiments and analyses are conducted to demonstrate the effectiveness and robustness of the proposed framework.

Weaknesses:

* The name GNNGuard is quite misleading since there is already a model that focuses on the robustness of GNNs named GNNGuard[1]. Therefore, changing another name could make it easier for readers to distinguish.

* There is another co-current work that also deals with the fingerprinting of GNNs[2]. Therefore the claims such as "To the best of our knowledge, no effort has been made to tailor the model fingerprinting for GNNs." should be reformat and include discussion and perhaps comparison with [2].


[1] GNNGuard: Defending Graph Neural Networks against Adversarial Attacks, NeurIPS 2020

[2] GrOVe: Ownership Verification of Graph Neural Networks using Embeddings, IEEE Symposium on Security and Privacy, 2024.

**Questions:**

Please kindly refer to the Weaknesses.

**Reviewer Confidence:**

2: The reviewer is willing to defend the evaluation, but it is likely that the reviewer did not understand parts of the paper

**Scope:**

4: The work is relevant to the Web and to the track, and is of broad interest to the community

---

### Decision · Program_Chairs · 2024-01-22

**Decision:**

Accept

**Comment:**

The setting is interesting and novel. One major concern seen among reviewers is to clarify the uniqueness of GNN architecture that requires novel methods to be developed for safeguarding the IP. Also the paper needs to emphasize that the method is specifically designed for graph learning frameworks.